

# Breaking away from the near horizon of extreme Kerr

**Alejandra Castro⋆ and Victor Godet†**

Institute for Theoretical Physics Amsterdam and Delta Institute for Theoretical Physics, University of Amsterdam, Science Park 904, 1098 XH Amsterdam, The Netherlands

⋆ a.castro@uva.nl, † V.Z.Godet@uva.nl

## Abstract

We study gravitational perturbations around the near horizon geometry of the (near) extreme Kerr black hole. By considering a consistent truncation for the metric fluctuations, we obtain a solution to the linearized Einstein equations. The dynamics is governed by two master fields which, in the context of the $nAdS_2/nCFT_1$ correspondence, are both irrelevant operators of conformal dimension $\Delta = 2$. These fields control the departure from extremality by breaking the conformal symmetry of the near horizon region. One of the master fields is tied to large diffeomorphisms of the near horizon, with its equations of motion compatible with a Schwarzian effective action. The other field is essential for a consistent description of the geometry away from the horizon.

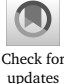

## 1 Introduction

Symmetries have played an important role in accounting for the quantum properties of black holes, and particularly the enhancement of symmetries that takes place for extremal and near-

extremal black holes [1–3]. The extremal limit of a black hole achieves zero Hawking temperature, even though the entropy remains finite and large. More prominently, it exhibits conformal invariance in the near horizon region and implies the existence of an $AdS_2$ factor [4–10]. Our understanding of (near-)extremal black holes is therefore tied to $AdS_2$ gravity, and our progress relies on our holographic understanding of this instance of AdS/CFT.

One the most infamous features of $AdS_2$ is that its symmetries do not allow for finite energy excitations [11, 12]. Dynamical processes force the introduction of a deformation away from $AdS_2$, and the duality that describes these deformations is known as the $nAdS_2/nCFT_1$ correspondence. This deformation is expected to be universal: breaking the conformal symmetry of $AdS_2$ induces a anomaly [13, 14] which governs the thermodynamic response and quantum chaos characterizing black holes. This expectation relies on studying 2D models of gravity coupled to a scalar field, colloquially referred to as Jackiw-Teitelboim (JT) gravity [15, 16]. In JT gravity, a non-trivial profile for the scalar field breaks explicitly conformal symmetry of $AdS_2$. The novelty is that this profile is tied to large diffeomorphisms at the boundary of $AdS_2$. These diffeomorphisms induce an anomaly via a Schwarzian derivative which governs the gravitational effects.

Reissner-Nordström black holes [17–20], with and without a cosmological constant, and the three dimensional BTZ solution [21, 22], fit well these advancements. In this context one can show that the dynamics of (near) extreme black holes is described by an effective theory of 2D gravity coupled to a scalar field. Other instances of this success include [23–32].

Rotating black holes add interesting features to this discussion. They share the $AdS_2$ factor, with the most prominent instance being the Near Horizon of Extreme Kerr (NHEK) in four dimensions [33]. A proposal for a holographic description of rotating black holes is the Kerr/CFT correspondence [34]; see [35] for a review of this program. They also share the dynamical obstructions that makes $AdS_2$ problematic [36, 37], which limits our holographic understanding. Recently, there has been some progress on quantifying rotating black holes along the lines of $nAdS_2/nCFT_1$ [38–40]. Rotation adds more complexity to the deformations, due to a squashing mode that breaks spherical symmetry. For certain 5D black holes it is possible to build a 2D model of gravity coupled to matter that encodes this complexity [39]. These models include non-trivial interactions that are not captured by JT gravity. Nevertheless the mechanism that breaks conformal symmetry for this example conforms with the thermodynamic response advocated in [13, 14].

Our goal here is to illustrate how to break the conformal symmetry of the near horizon geometry of the extreme Kerr solution. We will do this by solving the linearized Einstein equations around the near horizon geometry.[1] We are able to show that one of the gravitational perturbations incorporates a feature prominent in JT gravity: a scalar field that breaks conformal symmetry and is tied to the Schwarzian derivative. We also find an additional mode that is needed to consistently capture the deviations away from extremality, since its profile is non-vanishing for Kerr. We take this as evidence that simpler models, well suited for static black holes, do not accommodate rotating black holes.

---

[1]The study of gravitational perturbations of the Kerr black hole is extensive and impressive. We refer to [41] as a roadmap in this area. Examples of prior work on gravitational perturbations around NHEK that exploit its conformal symmetry are [42–44].

## 2  Near extreme Kerr

In this section we review properties of the near extreme Kerr geometry, with particular emphasis on its near horizon geometry. We start by considering the general Kerr solution,

$$ds^2 = -\frac{\Sigma\,\Delta}{(\tilde{r}^2 + a^2)^2 - \Delta\, a^2\sin^2\theta}d\tilde{t}^2 + \Sigma\left(\frac{d\tilde{r}^2}{\Delta} + d\theta^2\right)$$
$$+ \frac{\sin^2\theta}{\Sigma}((\tilde{r}^2 + a^2)^2 - \Delta\, a^2\sin^2\theta)\left(d\tilde{\phi} - \frac{2aM\tilde{r}}{(\tilde{r}^2 + a^2)^2 - \Delta\, a^2\sin^2\theta}d\tilde{t}\right)^2 , \qquad (1)$$

with

$$\Delta = (\tilde{r} - r_-)(\tilde{r} - r_+) , \quad r_\pm = M \pm \sqrt{M^2 - a^2} , \quad \Sigma = \tilde{r}^2 + a^2\cos^2\theta . \qquad (2)$$

Here $r_-$ and $r_+$ are the inner and outer horizons. We are using conventions where $G_4 = 1$. $M$ is the mass and $J = aM$ is the angular momentum of the black hole.

The extreme Kerr solution is obtained as the confluence of the inner and outer horizon: $r_+ = r_-$. We are interested in describing the dynamics of Kerr slightly above extremality. In this context, *near extremality* is defined as a deviation from the extreme limit which keeps $J$ fixed. Implementing it as a limit, we have

$$r_\pm = M_0 \pm \varepsilon\lambda + \frac{\varepsilon^2\lambda^2}{4M_0} + O(\lambda^3) , \qquad (3)$$

where $\lambda$ is a small parameter that controls deviations away from extremality. $M_0$ is the value of the mass at extremality, and $\varepsilon$ is a constant that controls the deviation of the mass above extremality. Under these conditions, we can identify a near horizon region. Redefining the coordinates in 1 as

$$\tilde{r} = \frac{r_+ + r_-}{2} + \lambda\left(r + \frac{\varepsilon^2}{4r}\right), \qquad \tilde{t} = 2M_0^2\frac{t}{\lambda} , \qquad \tilde{\phi} = \phi + M_0\frac{t}{\lambda} , \qquad (4)$$

and taking the limit $\lambda \to 0$ –with other parameters fixed– leads to the line element

$$ds^2 = M_0^2(1 + \cos^2\theta)\left[-r^2\left(1 - \frac{\varepsilon^2}{4r^2}\right)^2 dt^2 + \frac{dr^2}{r^2} + d\theta^2\right] \qquad (5)$$
$$+ M_0^2\frac{4\sin^2\theta}{1 + \cos^2\theta}\left[d\phi + r\left(1 + \frac{\varepsilon^2}{4r^2}\right)dt\right]^2 .$$

For $\varepsilon = 0$, this is the Near Horizon geometry of Extreme Kerr (NHEK) [33, 34]. For $\varepsilon \neq 0$, we will call this background the near-NHEK geometry.

It is instructive to discuss some properties of 5. For $\varepsilon = 0$, we have

$$ds^2 = M_0^2(1 + \cos^2\theta)\left(-r^2 dt^2 + \frac{dr^2}{r^2} + d\theta^2\right) + M_0^2\frac{4\sin^2\theta}{1 + \cos^2\theta}(d\phi + r\,dt)^2 . \qquad (6)$$

This geometry has four Killing vectors:

$$\xi_{-1} = \partial_t , \quad \xi_0 = t\partial_t - r\partial_r , \quad \xi_1 = \left(\frac{1}{r^2} + t^2\right)\partial_t - 2rt\partial_r - \frac{2}{r}\partial_\phi , \qquad k = \partial_\phi . \qquad (7)$$

These vectors generate an $sl(2) \times u(1)$ algebra which corresponds to the enhanced conformal symmetry of the near horizon geometry. One can also impose asymptotic boundary conditions

on 6. In particular, the set of diffeomorphisms preserving the asymptotic metric is [45]

$$t \longrightarrow f(t) + \frac{2f''(t)f'(t)^2}{4r^2 f'(t)^2 - f''(t)^2} \,,$$

$$r \longrightarrow \frac{4r^2 f'(t)^2 - f''(t)^2}{4r\, f'(t)^3} \,,$$

$$\phi \longrightarrow \phi + \log\left(\frac{2rf'(t) - f''(t)}{2rf'(t) + f''(t)}\right) \,, \tag{8}$$

where $f(t)$ is an arbitrary function that reflects the freedom of reparametrization the boundary metric.[2] Acting on 5, this diffeomorphism gives

$$ds^2 = M_0^2(1 + \cos^2\theta)\left[-r^2\left(1 + \frac{\{f(t),t\}}{2r^2}\right)^2 dt^2 + \frac{dr^2}{r^2} + d\theta^2\right] \tag{9}$$

$$+ \frac{4M_0^2 \sin^2\theta}{1 + \cos^2\theta}\left[d\phi + r\left(1 - \frac{\{f(t),t\}}{2r^2}\right)dt\right]^2 \,,$$

where

$$\{f(t),t\} = \left(\frac{f''}{f'}\right)' - \frac{1}{2}\left(\frac{f''}{f'}\right)^2 \,, \tag{10}$$

is the Schwarzian derivative. It is important to note that for $f(t) = e^{\varepsilon t}$, 9 reduces to the near-NHEK metric 5. At this stage, this implies that NHEK and near-NHEK are just one diffeomorphism away. It is also worth noting that the shift of $\phi$ in 8 is the large gauge transformation discussed in [46].

## 3  Gravitational perturbations

In this section we will study the response of NHEK to a small amount of energy: how the metric responds when we deviate from extremality. Our goal is to find a consistent truncation of the perturbations that captures the Schwarzian mode which is believed to be universal in the response to black hole near extremality. Our strategy is rather simple: we will propose an ansatz for the metric perturbations of NHEK and solve the linearized Einstein equations.

A deviation from extremality is a correction due to the near horizon parameter $\lambda$ introduced in 4. By inspection of the full on-shell Kerr geometry 1, which would correspond to stationary perturbations, it is clear that a suitable ansatz for metric perturbations needs to account for non-trivial $\theta$-dependence. With the insight on the behavior of Kerr, we will consider the following deviation of the NHEK geometry

$$ds^2 = -M_0^2 \frac{(1 + \cos^2\theta + \lambda\tilde{\chi}(t,r))}{1 + \lambda\psi(t,r)} r^2 dt^2 + M_0^2\left(1 + \cos^2\theta + \lambda\chi(t,r)\right)\left(\frac{dr^2}{r^2} + d\theta^2\right)$$

$$+ 4M_0^2 \frac{\sin^2\theta\,(1 + \lambda\Phi(t,r))}{1 + \cos^2\theta + \lambda\chi(t,r)}\,(d\phi + r\,dt + \lambda A)^2 \,, \tag{11}$$

where the one-form $A$ is supported in the $(t,r)$ subspace

$$A = A_t(t,r,\theta)dt + A_r(t,r,\theta)dr \,, \tag{12}$$

and captures the angular dependence of the ansatz. We treat the metric at linear order in $\lambda$. The metric perturbation $\Phi(t,r)$ parametrizes the change of the volume of the squashed

---

[2]Spoiler alert: this symmetry will be broken in the next section.

sphere; $\chi(t,r)$ characterizes the squashing parameter that breaks spherical symmetry; $\psi(t,r)$ and $\tilde{\chi}(t,r)$ are introduced for consistency of the ansatz. At this stage it is a guess that $\chi$, $\tilde{\chi}$ and $\psi$ have no $\theta$-dependence, and we will show that this is compatible with the equations of motion. We are not introducing $\phi$-dependence since it seems consistent, for the purpose of capturing deviations from extremality, to focus on solutions which respect the isometry due to the Killing vector $k = \partial_\phi$.

We now proceed to solve the linearized Einstein equations

$$R_{\mu\nu} = 0 \,, \tag{13}$$

where $R_{\mu\nu}$ is the 4D Ricci tensor, and look at the first correction due to $\lambda$ in 11. The $\theta$-components of this equation are the simplest to solve first. From $R_{t\theta}$ and $R_{\theta\phi}$ we can determine that the one-form can be written as

$$A = \alpha + \varepsilon_{ab}\partial^a \Psi \, dx^b \,, \qquad \Psi = \frac{1}{2\sin^2\theta}\left[\left(1 + \frac{\sin^4\theta}{4}\right)\Phi(t,r) - \chi(t,r)\right] \,, \tag{14}$$

with

$$\alpha = \alpha_t(t,r,\theta)dt + \alpha_r(t,r)dr \,, \qquad \alpha_t(t,r,\theta) = a_1(t,r) + a_2(r,\theta) \,. \tag{15}$$

The components of $\alpha$ are arbitrary functions at this stage. In 14 we introduced an auxiliary 2D metric, defined as

$$\gamma_{ab}dx^a dx^b = -r^2 dt^2 + \frac{dr^2}{r^2} \,, \tag{16}$$

and $\varepsilon_{ab}$ is the Levi-Civita tensor of this space, with $\varepsilon_{tr} = \sqrt{-\det\gamma_{ab}}$. This is the AdS$_2$ space appearing in the NHEK geometry 6. Using 14 in $R_{r\theta}$ and $R_{\theta\theta}$, we can see that $a_2 = 0$, and that $\tilde{\chi} = \chi$. In addition $R_{\theta\theta} = 0$ implies

$$\Box_2 \chi = 2\chi \,, \tag{17}$$

where $\Box_2$ is the Laplacian for the AdS$_2$ background 16, and therefore $\chi$ is an operator of conformal dimension $\Delta = 2$. With this input in place, setting $R_{\phi\phi} = 0$ leads to

$$\psi(t,r) = -\Phi + \Box_2\Phi - 2\,\varepsilon^{ab}\partial_a \alpha_b \,. \tag{18}$$

We have five components left to solve: $R_{tt}$, $R_{tr}$, $R_{t\phi}$, $R_{rr}$ and $R_{r\phi}$. Using the previous equations, one of these components is redundant. After some simple manipulations, we find

$$\Phi(t,r) = \Phi_0 + \Phi_{\mathrm{JT}}(t,r) \,. \tag{19}$$

Here $\Phi_0$ is a constant: this is the degree of freedom that changes the value of $M_0$, since it can be reabsorbed as a rescaling of the angle $\phi$. The field $\Phi_{\mathrm{JT}}$ satisfies

$$\nabla_a \nabla_b \Phi_{\mathrm{JT}} - \gamma_{ab}\Box_2\Phi_{\mathrm{JT}} + \gamma_{ab}\Phi_{\mathrm{JT}} = 0 \,, \tag{20}$$

which is the equation of motion of the scalar field in Jackiw-Teitelboim gravity [15, 16]; see Appendix A for a brief review. Finally, we also have

$$\alpha = -\varepsilon_{tr}\partial^t \Phi \, dr + \tilde{\alpha} \,. \tag{21}$$

There is also a constraint on $\tilde{\alpha}$, but this makes it pure gauge: we can remove $\tilde{\alpha}$ via a trivial diffeomorphism. The details are given in Appendix B.

In summary, the linearized perturbations are captured by two fields: $\chi$ and $\Phi$. By solving the dynamics of these two fields, dictated by 17 and 20 one can reconstruct consistently the metric near NHEK. At this stage it is important to make some technical remarks:

1. Our analysis is also a consistent truncation of the linearized Einstein equations around the locally NHEK background 9 where we take the ansatz for the perturbations to have the same form as in 11. The explicit form of the perturbed metric can be found in 47. The solution is given by 14-21, with the modification that the auxiliary 2D metric in 16 is changed to a locally AdS$_2$ metric:[3]

$$\gamma_{ab} \mathrm{d}x^a \mathrm{d}x^b = -r^2 \left(1 + \frac{\{f(t), t\}}{2r^2}\right)^2 \mathrm{d}t^2 + \frac{\mathrm{d}r^2}{r^2} \,. \tag{22}$$

In particular, the solutions to 20 on this background are of the form

$$\Phi_{\mathrm{JT}} = \nu(t)\, r + \frac{\mu(t)}{r} \,, \tag{23}$$

where $\nu$ obeys

$$\left(\frac{1}{f'}\left(\frac{(f'\nu)'}{f'}\right)'\right)' = 0 \,, \tag{24}$$

and $\mu$ satisfies 33. This equation relates the explicit breaking of symmetries in NHEK, due to $\nu(t)$, with the diffeomorphism 8 on its boundary, parametrized by $f(t)$. It can also be obtained from the Schwarzian effective action 35, as reviewed in Appendix A. See [14] for more details on this relation and its interpretation. In Appendix C, we show how to obtain the Schwarzian action for near-NHEK from the 4D Einstein-Hilbert action. We also reproduce the linear temperature response in the entropy of the near-extremal Kerr solution as expected from the general arguments in [14].

2. We constructed a consistent truncation of the linearized problem that captures the deviations away from the AdS$_2$ throat of the extremal Kerr solution. We do not expect 11 to be the most general ansatz for gravitational dynamics near the NHEK geometry: additional angular dependence could be added, which will be interesting to quantify. In particular, it would be interesting to develop a more systematic construction of master fields along the lines of the techniques developed by Kodama-Ishibashi [47,48], and the recent results in [49].

3. It is instructive to match the perturbations derived in this section with the stationary configuration that would match the behavior of the Kerr black hole. Applying the limit 4 to the Kerr geometry 1, and comparing the linear order in $\lambda$ with the perturbations 11 for near-NHEK, we obtain

$$\chi_{\mathrm{Kerr}} = \Phi_{\mathrm{Kerr}} = \frac{2}{M_0}\left(r + \frac{\varepsilon^2}{4r}\right) \,, \tag{25}$$

and the one-form $\alpha$ is zero. Hence both modes are non-trivial for the Kerr solution.

The nAdS$_2$ analysis of the Kerr black hole shares one similarity with the charged counterparts studied in [17, 18]: there is one gravitational mode $\Phi$ which satisfies the JT equations of motion 20. For Reissner-Nordström black holes, it was consistent to only focus on the dynamics of $\Phi$ as the leading effect in deviations away from extremality. But there are some important differences for Kerr. First, the $\theta$-dependence in 14 prevents us from building a 2D effective theory that describes these modes. This is mostly a technical barrier, since it is more cumbersome to keep track of the dynamics of the system. Nonetheless, we expect to be able

---

[3]Although the formula 21 is not covariant with respect to the 2D metric $\gamma_{ab}$, it still holds for a linearized perturbation around near-NHEK.

to quantify, for example, correlation functions of these gravitational perturbations in future work.

The second, and most important, difference relative to Reissner-Nordström black holes is the additional degree of freedom $\chi$ that we have found. This is similar to the 5D rotating black holes studied in [39]: there is a squashing mode $\chi$ that influences the gravitational perturbations. Remarkably, $\chi$ and $\Phi$ are both irrelevant operators of conformal dimension $\Delta = 2$. While the dynamics of $\Phi$ is restricted by the large diffeomorphism of NHEK, via 24, the field $\chi$ is a dynamical mode. As indicated by 25, the source for $\chi$ is turned on for the Kerr solution: this a strong indication that although 24 captures some important aspects of the deviations away from extremality, a complete characterization needs to take into account the interactions of $\Phi$ with $\chi$.

Large diffeomorphisms play a prominent role in our analysis, which begs for a comparison with Kerr/CFT. A crucial difference is that the asymptotic symmetry group used in [34] had arbitrary functions of $\phi$, while here we are considering generators that reparametrize the boundary time.[4] It would be interesting to investigate whether there is a deformation of NHEK that ties the explicit breaking of the conformal symmetry by an irrelevant deformation to the conformal anomaly in the Virasoro algebra of Kerr/CFT. This will require searching for gravitational perturbations that have non-trivial $\phi$-dependence, which we have ignored in this work. We hope to pursue this direction in future work.

## Acknowledgements

We are grateful to Shahar Hadar, Jorrit Kruthoff, Achilleas Porfyriadis, Joan Simon, and Wei Song for discussions on this topic. AC would like to thank the participants of Lorentz Center workshop "Singularities and Horizons: From Black Holes to Cosmology" for useful discussions. This work is supported by Nederlandse Organisatie voor Wetenschappelijk Onderzoek (NWO) via a Vidi grant, and by the Delta ITP consortium, a program of the NWO that is funded by the Dutch Ministry of Education, Culture and Science (OCW).

## A   Aspects of JT gravity

In this appendix we review some basic properties of JT gravity [15, 16]; our summary is based on [13, 14, 51]. The 2D JT action with a negative cosmological constant is given by

$$I_{2D} = \frac{1}{16\pi G_2} \int d^2x \sqrt{-g}\, \Phi\, (R+2) + \frac{1}{8\pi G_2} \int dt \sqrt{-h}\, \Phi K \,. \tag{26}$$

The on-shell metrics are all locally AdS$_2$. The equation of motion for $\Phi$ takes the form

$$\nabla_a \nabla_b \Phi - g_{ab} \Box_2 \Phi + g_{ab} \Phi = 0 \,. \tag{27}$$

For AdS$_2$ in the coordinates used in 16, the explicit solution is

$$\Phi(t,r) = c_1 r + c_2\, r t + c_3 \left( r t^2 - \frac{1}{r} \right) \,, \tag{28}$$

where $c_1, c_2$ and $c_3$ are arbitrary constants.

---

[4]In the context of Kerr/CFT, our symmetry group follows more closely the analysis in [50].

Next, consider a diffeomorphism that preserves the boundary of AdS$_2$ and the radial gauge

$$t \longrightarrow f(t) + \frac{2f''(t)f'(t)^2}{4r^2 f'(t)^2 - f''(t)^2} \,,$$
$$r \longrightarrow \frac{4r^2 f'(t)^2 - f''(t)^2}{4r\, f'(t)^3} \,, \tag{29}$$

which is the 2D version of 8. The metric transforms as

$$ds^2 \;=\; -r^2 \left( 1 + \frac{s(t)}{2r^2} \right)^2 dt^2 + \frac{dr^2}{r^2} \,, \tag{30}$$

where

$$s(t) \equiv \{f(t), t\} = \left( \frac{f''}{f'} \right)' - \frac{1}{2} \left( \frac{f''}{f'} \right)^2 \,. \tag{31}$$

The solution for $\Phi$ is now modified to

$$\Phi = \nu(t)\, r + \frac{\mu(t)}{r} \,, \tag{32}$$

where

$$\mu' = \frac{s(t)}{2} \nu' \,, \quad 2\mu + s(t)\nu + \nu'' = 0 \,. \tag{33}$$

Combining them gives

$$\left( \frac{1}{f'} \left( \frac{(f'\nu)'}{f'} \right)' \right)' = 0 \,. \tag{34}$$

This last equation relates dynamically the source in $\Phi$ to the diffeomorphism 29 that induces a reparametrization of the boundary. Although the relation 34 is derived from the 2D equations of motion, it can also be captured by a 1D boundary action

$$I_{\text{bndy}} = \frac{1}{8\pi G_2} \int dt \, \nu(t)\{f(t), t\} \,, \tag{35}$$

which is the Schwarzian effective action. $I_{\text{bndy}}$ is obtained by evaluating 26 for locally AdS$_2$ metrics 30, and focusing on the finite terms near the boundary. The variation of $I_{\text{bndy}}$ with respect to $f$ gives 34.

## B    Redundancies due to diffeomorphisms

In this appendix we determine which components of the metric fluctuations in 11 correspond to pure diffeomorphisms. First consider an arbitrary infinitesimal diffeomorphism

$$\delta x^\mu = \xi^\mu(t, r, \theta, \phi), \tag{36}$$

which leads to a perturbation

$$\delta g_{\mu\nu} = \mathcal{L}_\xi g_{\mu\nu} \,, \tag{37}$$

where $g_{\mu\nu}$ is the NHEK metric 6. Demanding that the perturbation $\delta g_{\mu\nu}$ fits in the ansatz 11 gives some constraints on $\xi^\mu$ which can be solved explicitly. From this analysis, we can show that $\Phi$ and $\chi$ are physical fields and that the one-form $\tilde{\alpha}$ is pure gauge.

To see that $\tilde{\alpha}$ can be removed by a diffeomorphism, we first need to solve the following constraint which comes from the $(t,t)$ component of the linearized Einstein equation. Using 14-20 on $R_{tt} = 0$ gives[5]

$$\partial_r \left( r^3 \partial_r (\partial_t \tilde{\alpha}_r - \partial_r \tilde{\alpha}_t) \right) = 0 \,. \tag{38}$$

This constraint can be integrated explicitly and we can write the result as follows

$$\tilde{\alpha}_r(t,r) = \partial_r F(t,r), \tag{39}$$
$$\tilde{\alpha}_t(t,r) = \partial_t F(t,r) + \frac{G^{(3)}(t)}{2r} + H'(t)r \,,$$

where $F(t,r), G(t)$ and $H(t)$ are arbitrary functions. The infinitesimal diffeomorphism that we are looking for is then given by

$$\xi = \left( -H + G(t) + \frac{G''(t)}{2r^2} \right) \partial_t - rG'(t)\partial_r - \left( F(t,r) + G''(t) \right) \partial_\phi \,. \tag{40}$$

Indeed, the corresponding perturbation takes the form

$$\mathcal{L}_\xi g = 2M_0^2(1 + \cos^2\theta)(\partial_t \tilde{\alpha}_r - \partial_r \tilde{\alpha}_t) r^2 \mathrm{d}t^2 + \frac{8M_0^2 \sin^2\theta}{1 + \cos^2\theta}(\tilde{\alpha}_t \mathrm{d}t + \tilde{\alpha}_r \mathrm{d}r)(\mathrm{d}\phi + r\mathrm{d}t) \,, \tag{41}$$

and precisely cancels the contribution of $\tilde{\alpha}$ in the solution of our ansatz 11. We have also noticed that the perturbations associated with the gravitational mode $\Phi$ are related to some large diffeomorphisms of the NHEK with non-trivial $\phi$-dependence. We hope to investigate them in future work.

## C  On-shell action and thermodynamics

It is instructive to discuss the thermodynamics near extremality, and its ties to the gravitational perturbation $\Phi$. The thermodynamic properties of the near-NHEK geometry are as follows [52]: implementing 3 on the standard thermodynamic variables, the energy above extremality is

$$E = M - M_0 = \frac{\varepsilon^2 \lambda^2}{4M_0} + O(\lambda^3) \,. \tag{42}$$

The near-extremal entropy at linear order in $\lambda$ is

$$S_{\mathrm{BH}} = \frac{A_H}{4} = 2\pi M_0^2 + 2\pi M_0 \varepsilon \lambda + O(\lambda^2) \,, \tag{43}$$

and in this limit the Hawking temperature is given by

$$T = \frac{r_+ - r_-}{8\pi M r_+} = \frac{\varepsilon \lambda}{4\pi M_0^2} + O(\lambda^2) \,. \tag{44}$$

This allows us to write

$$E = CT^2 + O(T^3) \,, \qquad S = 2\pi M_0^2 + 2CT + O(T^2) \,, \tag{45}$$

where $C = 4\pi^2 M_0^3$.

---

[5]Solving $R_{rr} = 0$ gives the same constraint as $R_{tt} = 0$ after using 14-20.

We will see that these thermodynamical properties can be understood using the renormalized on-shell action, along the lines of [14]. Let's consider

$$I_{4D} = \frac{1}{16\pi} \int_M d^4x \sqrt{|g|} R + \frac{1}{8\pi} \int_{\partial M} d^3x \sqrt{|h|} K \,, \tag{46}$$

which is the standard Einstein-Hilbert action with the addition of the Gibbons-Hawking-York term. We would like to evaluate $I_{4D}$ on the general perturbation of the locally NHEK background. The on-shell solution is

$$
\begin{aligned}
ds^2 = & -M_0^2 \frac{(1 + \cos^2\theta + \lambda \tilde{\chi}(t,r))}{1 + \lambda \psi(t,r)} r^2 \left( 1 + \frac{\{f(t),t\}}{2r^2} \right)^2 dt^2 \\
& + M_0^2 (1 + \cos^2\theta + \lambda \chi(t,r)) \left( \frac{dr^2}{r^2} + d\theta^2 \right) \\
& + 4M_0^2 \frac{\sin^2\theta \, (1 + \lambda \Phi(t,r))}{1 + \cos^2\theta + \lambda \chi(t,r)} \left( d\phi + r \left( 1 - \frac{\{f(t),t\}}{2r^2} \right) dt + \lambda A \right)^2 \,,
\end{aligned}
\tag{47}
$$

which we treat at linear order in $\lambda$, and the fields obey 14-21 with background metric 22. Replacing 47 in the 4D action 46 leads to divergences that are common for on-shell gravitational actions. To remove them, we will take a standard route: after specifying a set of boundary conditions, we will build a renormalized action by requiring that its variation is finite. Our setup follows closely the rules of holographic renormalization in AdS gravity, with [39] being the closest example, and any deviation from these rules will be highlighted.

To start, it is convenient to rewrite 47 as an asymptotic solution with arbitrary sources for the fields:

$$
\begin{aligned}
ds^2 = & M_0^2 \frac{(1 + \cos^2\theta + \lambda \tilde{\chi}(t,r))}{1 + \lambda \psi(t,r)} \gamma_{tt}(t,r) dt^2 \\
& + M_0^2 (1 + \cos^2\theta + \lambda \chi(t,r)) \left( \frac{dr^2}{r^2} + d\theta^2 \right) \\
& + 4M_0^2 \frac{\sin^2\theta \, (1 + \lambda \Phi(t,r))}{1 + \cos^2\theta + \lambda \chi(t,r)} \left( d\phi + a_t(t,r) dt + \lambda A \right)^2 \,.
\end{aligned}
\tag{48}
$$

For $\tilde{\chi}$, $\psi$, and $A$ we will be using the on-shell values determined by $\gamma_{tt}$, $\Phi$ and $\chi$ as described in Section 3. For the additional fields, we have

$$
\sqrt{-\gamma_{tt}} = \alpha(t) r + \frac{\beta(t)}{r} \,, \qquad\qquad a_t = \alpha(t) r - \frac{\beta(t)}{r} + \zeta(t) \,, \tag{49}
$$
$$
\Phi = \nu(t) r + \frac{\mu(t)}{r} \,, \qquad\qquad \chi = \sigma(t) r + \cdots + \frac{\kappa(t)}{r^2} + \cdots \,.
$$

Here we identify $\alpha$, $\nu$, $\sigma$ as sources for $\gamma_{tt}$, $\Phi$ and $\chi$, respectively; the functions $\beta$, $\mu$ and $\kappa$ are the corresponding vevs. $\zeta$ is the source for $a_t$, while its charge is one in our conventions.[6] Note that for $\chi$ we are only highlighting its source and vev: the dots are subleading terms in the large $r$ expansion that are determined by imposing its equation of motion. In this notation, the solution to equation 33 reads

$$
\beta(t) = \frac{\alpha(t) \mu'(t)}{\nu'(t)} \,, \qquad\qquad \mu(t) = \frac{c_0}{\nu(t)} - \frac{\nu'(t)^2}{4\alpha(t)^2 \nu(t)} \,, \tag{50}
$$

where $c_0$ is a constant.

---

[6]For a 2D Maxwell field we are simply identifying the electric charge $Q$ from $F_{rt} = Q\sqrt{|\gamma|}$.

The renormalized action is of the form

$$I_{\text{ren}} = I_{4D} + I_{ct} \,, \tag{51}$$

where $I_{4D}$ is specified above and $I_{ct}$ is a counterterm action. We want to cast our variational problem with respect to the 2D variables in 49. Leaving the gauge field fixed, for reasons explained below, we set up the variation of the action as follows:

$$\begin{aligned}
\delta I_{\text{ren}} &= \int_\Sigma d^3x \, \pi^{\mu\nu} \delta h_{\mu\nu} \\
&= \int_\Sigma d^3x \left( \Pi_\Phi \delta\Phi + \Pi^{tt} \delta\gamma_{tt} + \Pi_\chi \delta\chi \right) \\
&= \int dt \left( \pi_\alpha \delta\alpha(t) + \pi_\nu \delta\nu(t) + \pi_\sigma \delta\sigma(t) \right) \,,
\end{aligned} \tag{52}$$

where $\Sigma$ is a cutoff surface of constant $r$ with induced metric $h_{\mu\nu}$. From the first to the second line we are simply casting the variation of the 3D boundary metric $h_{\mu\nu}$ in terms of the 2D fields. In the last line we are specifying the variations of the 2D fields in terms of their sources, and we have integrated over the angular variables $(\theta, \phi)$. Fixing the variation of the gauge field in this notation means that we do not vary the sources appearing in $a_t$ and $A$. The task is now to build $I_{ct}$ such that the momenta $\pi_\alpha$, $\pi_\nu$, and $\pi_\sigma$ are finite as we approach the boundary at $r \to \infty$.

In terms of the 3D variables, the momenta $\pi^{\mu\nu}$ receives a contribution from $I_{4D}$ which is the usual Brown-York stress tensor:

$$\pi^{\mu\nu}_{4D} = \frac{\delta I_{4D}}{\delta h_{\mu\nu}} = -\frac{1}{16\pi} \sqrt{-h} \left( K^{\mu\nu} - K h^{\mu\nu} \right) \,. \tag{53}$$

This term will lead to divergences in $\pi_\alpha$, $\pi_\nu$, and $\pi_\sigma$ as we take $r \to \infty$; in particular we get

$$\begin{aligned}
\pi_{\alpha,4D} &= \frac{M_0^2}{2} \left( \nu(t) r^2 - \mu(t) \right) \lambda - \frac{M_0^2}{8} \nu(t) (4\nu(t) - \pi\sigma(t)) \lambda^2 r^3 + \cdots \\
\pi_{\nu,4D} &= \frac{M_0^2}{2} \left( \alpha(t) r^2 - \beta(t) \right) \lambda - \frac{M_0^2}{8} \alpha(t) (2\nu(t) - (\pi-2)\sigma(t)) \lambda^2 r^3 + \cdots \\
\pi_{\sigma,4D} &= \frac{M_0^2}{32} \alpha(t) (4(\pi-2)\nu(t) - (4+3\pi)\sigma(t)) \lambda^2 r^3 + \cdots \,,
\end{aligned} \tag{54}$$

where the dots are higher-order terms in $\lambda r$, and we have integrated over the angular variables $(\theta, \phi)$. It is important to emphasize that our perturbative expansion is only meaningful at leading order in the deformations we turn on, which implies that $\lambda r \ll 1$ as $r \to \infty$.

The leading divergences in the canonical momenta $\pi_\alpha$, $\pi_\nu$ and $\pi_\sigma$ can be cancelled using the following counterterms

$$I_{ct} = \frac{M_0^2}{8} \int dt \sqrt{-\gamma_{tt}} \left( c_1 \lambda\Phi + c_2 \lambda^2\Phi^2 + c_3 \lambda^2\chi^2 + c_4 \lambda^2\Phi\chi \right) \,, \tag{55}$$

where the coefficients are found to be

$$c_1 = -4, \qquad c_2 = 1, \tag{56}$$

$$c_3 = \frac{1}{8}(4+3\pi), \qquad c_4 = 2 - \pi \,.$$

Note that the counterterms used here are very similar to those in [39] which also displays similar equations of motion. Adding the contribution from these counterterms to 54, the renormalized momenta are

$$\pi_\alpha = \pi_{\alpha,\text{4D}} + \pi_{\alpha,\text{ct}} = -M_0^2\,\mu(t)\,\lambda + O(\lambda^2 r)\,,$$

$$\pi_\nu = \pi_{\nu,\text{4D}} + \pi_{\nu,\text{ct}} = -M_0^2\,\beta(t)\,\lambda + \frac{3M_0^2}{4}\alpha(t)\kappa(t)\,\lambda^2 + O(\lambda^2 r)\,,$$

$$\pi_\sigma = \pi_{\sigma,\text{4D}} + \pi_{\sigma,\text{ct}} = \frac{3M_0^2}{32}(\pi+4)\,\alpha(t)\kappa(t)\,\lambda^2 + O(\lambda^2 r)\,. \tag{57}$$

We have retained some subleading terms in conformal perturbation theory: this is to illustrate the different behavior of $\chi$ compared to $\Phi$. Because the momenta for $\Phi$ is influenced by the large diffeormorphism of the background metric, the finite contribution appears at $O(\lambda)$. In constrast, $\chi$ behaves as a more traditional propagating field in AdS, and hence the term $\kappa(t)\delta\sigma(t)$ appears at $O(\lambda^2)$.

Using 57 in 52, the renormalized variation is

$$\delta I_{\text{ren}} = -M_0^2\lambda \int dt\,(\mu(t)\delta\alpha(t) + \beta(t)\delta\nu(t)) + O(\lambda^2)\,, \tag{58}$$

which can be integrated using the relations 50 and evaluated on-shell to give the effective action

$$I_{\text{ren}} = -\frac{M_0^2\lambda}{2} \int dt\left(\nu(t)\{f(t),t\} + \frac{4c_0}{\nu(t)}\right) + O(\lambda^2)\,. \tag{59}$$

We can compare with the near-extremal entropy by evaluating this action on the near-extremal black hole. Using 5 and 25 we have

$$\{f(t),t\} = -\frac{\varepsilon^2}{2}\,, \qquad \nu(t) = \frac{2}{M_0}\,, \qquad c_0 = 0\,. \tag{60}$$

Going to Euclidean signature by taking $t \to -it_E$, we can derive the near-extremal entropy from the Euclidean renormalized action $I_E = -iI_{\text{ren}}$ on a circle of size $2\pi/\varepsilon$ according to

$$\delta S_{\text{BH}} = (1 + \varepsilon\partial_\varepsilon)(-I_E) = 2\pi M_0\varepsilon\lambda\,. \tag{61}$$

This matches the linear response of the thermodynamics in 43.

Finally, we return to the role of the gauge field in our variational problem. The treatment of this field is more delicate since the source $\zeta(t)$ in 49 is subleading compared to its electric charge and the backreaction in 14. This is a known effect in 2D theories with a Maxwell field, and how to properly treat this is discussed in detail in [21,39]. Following that discussion, one simple way to circumvent the issues related to the gauge field is to freeze it in the variational problem, and focus on the remaining variables. This would not be the most general variational problem, but it suffices to capture the Schwarzian effective action as illustrated by our computations.

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
