# Peer review of "Breaking away from the near horizon of extreme Kerr"

_SciPost Physics, doi:SciPost Phys. 8, 089 (2020)_

## Round 2 · Referee Report · Anonymous (Referee 1) · 2019-9-26

Strengths

1-The paper is well-written and pointed.

2-The relation between linearized perturbations of NHEK and the Jackiw-Teitelboim model is interesting.

3-The physical interpretation of the two fields governing the linearized perturbations, χ and Φ, is intriguing.

Weaknesses

1-The ansatz for linearized perturbations (3.1) does not capture the most general perturbations possible.

2-No attempt was made to holographically renormalize the action (c.7), which made it impossible to derive the free energy from an on-shell action.

3-For this rather short paper there seem to be quite a few loose ends that could be tied up with a little bit of extra effort. In particular, points 1 and 2 above could be addressed by doing straightforward calculations. I did not find good reasons in the paper why this was not done.

Report

The paper focuses on a specific class of linearized gravitational perturbations away from the NHEK geometry.

After a concise review of the NHEK geometry near-NHEK is presented, which is identified as being diffeomorphic to NHEK, with a diffeomorphism involving the Schwarzian derivative. In the main section 3 an ansatz is provided for gravitational perturbations that allows to deduce interesting consequences. While there are some justifications for the ansatz given in the paper, it is not a generic ansatz (as mentioned in the concluding statements). A key result is that the specific ansatz, even though it involves six free functions, some of which depend on the angle θ, is completely characterized only by two arbitrary functions of time and radius. One of the fields is shown to obey the same equations as the dilaton field in the Jackiw-Teitelboim model. The other field solves a Klein-Gordon type of equation on an AdS2 background, thus allowing them to interpret it as conformal primary. From a CFT viewpoint both fields correspond to irrelevant operators (of weight 2).

The appearance of the Schwarzian derivative in the perturbed AdS2 metric allows to connect their analysis to the gravity side of SYK and shows that the conformal symmetry breaking of NHEK to near-NHEK resembles the nAdS2/CFT1 correspondence, which is one of their main conclusions.

Requested changes

1-For clarity I suggest to replace in the first paragraph of the introduction "Our understanding of black holes is therefore tied to AdS2 gravity" by "Our understanding of (near-)extremal black holes is therefore tied to AdS2 gravity".

2-I do not quite see the point in providing an incomplete calculation, even if it is in an appendix. One possibility is to delete the part of appendix C that starts two lines above (C.7) and just add the statement that it would be interesting to derive the free energy from an (appropriately holographically renormalized) on-shell action. A better possibility would be to holographically renormalize the action by adding suitable boundary terms and to derive the free energy.

3-I am aware that this would go beyond what the authors envisaged, but most likely it would be worthwhile to pursue the generalization of the perturbative analysis envisaged by the authors in their last paragraph of section 3. Since the first submission of the paper to the arXiv was already more than 3 months ago perhaps they have already partial results that they could wrap up soon and include in an enhanced version of the paper. If not, the paper still would be suitable for publication once the two points above are addressed.

  • validity: high
  • significance: good
  • originality: good
  • clarity: top
  • formatting: perfect
  • grammar: excellent

Author:  Alejandra Castro  on 2019-12-07  [id 668]

(in reply to Report 1 on 2019-09-26)
Category:
answer to question
correction

We would like to thank the referee for the positive and constructive remarks regarding our manuscript.

In regards to the requested changes, our response is the following:

  1. We changed the sentence as suggested by the referee.

  2. We re-did App. C using holographic renormalization techniques, and we now obtain the Schwarzian effective action. We can account for the linear temperature response of the entropy near-extremality from the Schwarzian theory, as it is universally expected. One of the reasons the first attempt (prior version of App C) didn't work is due to the backreaction of the 2D metric, and the asymptotic behaviour of the 2D gauge field. In the current version of this appendix we managed to address these issues in a more systematic way. Version 3 in the arxiv explains our new approach, and we are obtaining reasonable answers.

  3. This is a very important aspect of the program we are developing for 4D Kerr. We are currently working on this aspect with other collaborators, and we are confident that within the next year we will have a complete analysis. This project is a complex and lengthy analysis of the Teukolsky formalism for Kerr, since it will also include the matching problem with gravitational modes in the full Kerr geometry. The results we have so far are consistent with the results here, but it is not a short addition and it deserves a separate publication. We did not modify the manuscript in this regard.

We hope our answers addresses the concerns of the referee. And we are happy to discuss further any questions that might arise.

---

## Editorial Decision

published